# The Occurrence of Freshwater Fish-Borne Zoonotic Helminths in Italy and Neighbouring Countries: A Systematic Review

**DOI:** 10.3390/ani13243793

**Published:** 2023-12-08

**Authors:** Vasco Menconi, Elena Lazzaro, Michela Bertola, Lisa Guardone, Matteo Mazzucato, Marino Prearo, Ewa Bilska-Zajac, Luana Cortinovis, Amedeo Manfrin, Giuseppe Arcangeli, Giorgia Angeloni

**Affiliations:** 1Istituto Zooprofilattico Sperimentale delle Venezie, Viale dell’Università, 10, 35020 Legnaro, Italy; vmenconi@izsvenezie.it (V.M.); elazzaro@izsvenezie.it (E.L.); mmazzucato@izsvenezie.it (M.M.); lcortinovis@izsvenezie.it (L.C.); amanfrin@izsvenezie.it (A.M.); garcangeli@izsvenezie.it (G.A.); gangeloni@izsvenezie.it (G.A.); 2Istituto Zooprofilattico Sperimentale del Piemonte Liguria e Valle D’Aosta, Via Bologna 148, 10154 Torino, Italymarino.prearo@izsto.it (M.P.); 3Department of Parasitology and Invasive Diseases, National Veterinary Research Institute, Partyzantow Avenue 57, 24-100 Pulawy, Poland; ewa.bilska@piwet.pulawi.pl

**Keywords:** fish borne zoonotic helminths, *Contracaecum rudolphii*, *Dibothriocephalus latus*, *Eustrongylides excisus*, *Opisthorchis felineus*, *Clinostomum complanatum*, *Pseudamphistomum truncatum*

## Abstract

**Simple Summary:**

This systematic review focuses on the occurrence of fish-borne zoonotic helminths, including *Clinostomum complanatum*, *Contracaecum rudolphii*, *Dibothriocephalus latus*, *Eustrongylides excisus*, *Opisthorchis felineus*, and *Pseudamphistomum truncatum*, in freshwater fish populations of Italy and neighbouring countries. The study outlines the biological aspects and investigates the factors involved in the geographical expansion of these parasitic species. By synthesizing existing knowledge, we aim to compile epidemiological information concerning fish-borne zoonotic helminths and highlight the consumer risks. In conclusion, we encourage a One-Health approach in the context of food safety among EU countries to manage sanitary issues of all fish-borne zoonoses.

**Abstract:**

In recent years, the consumption of fish products has surged in European countries, being an essential part of a healthy diet. Despite representing a small part of EU production, freshwater fisheries hold considerable significance for lake-dwelling populations and tourists seeking traditional dishes. This increased fish consumption has brought to light potential health risks associated with fish-borne zoonotic helminths (FBZHs), now acknowledged as global food-borne parasites. Fish-borne zoonotic helminths belong to various taxonomic groups, including nematodes (Anisakidae), trematodes (Opisthorchiidae and Heterophyidae), and cestodes (Diphyllobothriidae). More than 50 species of FBZH are known to cause human infections, derived from eating raw or undercooked aquatic foods containing viable parasites. Despite increased attention, FBZHs remain relatively neglected compared to other food-borne pathogens due to factors like chronic disease progression and under-diagnosis. This systematic review concentrates on the prevalence of six freshwater FBZHs (*Clinostomum complanatum*, *Contracaecum rudolphii*, *Dibothriocephalus latus*, *Eustrongylides excisus*, *Opisthorchis felineus*, and *Pseudamphistomum truncatum*) in Italy and neighbouring countries. The study explores the expansion of these parasites, analysing their biological and epidemiological aspects, and the factors that influence their proliferation, such as the increased cormorant population and the lake eutrophication phenomena. In summary, this research highlights the necessity for further research, the development of spatial databases, and the establishment of a unified European policy to effectively manage these multifaceted health concerns. It strongly advocates adopting a One-Health approach to address the growing incidence of parasitic zoonoses within the context of food safety in EU countries.

## 1. Introduction

In recent years, fish products have become an increasingly popular source of protein as an essential part of a healthy diet. In European countries (EU), the consumption of fish products per capita reached up to 24 Kg and about 1200 fish species have been commercialized. In 2021, extra-EU imports of fishery and aquaculture products totalled 6.23 million tonnes. Spain is in first place with 1.17 million tons (18.9%) and Italy in sixth place with 453,000 tons (7.3%) [1,2,3,4]. Freshwater fish products represent a small part of EU production; however, they have local commercial importance for the inhabitants of lake areas and for tourists attracted by traditional dishes. Various relevant fish species, carrying zoonotic parasites, could pose a risk for public health. In particular, fish-borne zoonotic helminths (FBZHs) are listed among the top food-borne parasites at the global level [4,5]. Fish-borne zoonotic helminths belong to various taxonomic groups, including nematodes (Anisakidae), trematodes (Opisthorchiidae and Heterophyidae), and cestodes (Dibothriocephalidae) [5]. More than 50 species of FBZH are known to cause human infections, derived from eating raw or undercooked aquatic foods containing viable parasites. Data on fish parasitic zoonoses have been initially limited to the populations living along coastal and lacustrine regions from low/middle-income countries, particularly in Asian countries where large quantities of raw fish products are traditionally consumed [6,7]. The latest epidemiological updates provide a new scenario characterized by both an increase in human cases and an expansion of the geographical distribution of parasitosis at a global level [6,8,9,10]. Increasing the global food trade, demographic changes, new culinary habits, and climate change are among the main factors influencing the spread of FBZHs [6,11]. In fact, the globalization of the food supply has favoured the trade of a wide variety of fish products worldwide [12]. In particular, the importation of fish products from countries which do not have well-developed food control systems could be a considerable risk factor for FBZHs spreading worldwide. International mobility facilitates the translocation of infectious agents from one country to another, resulting in the spreading of parasites to areas outside endemic zones [13,14]. Moreover, the rising interest of European consumers in exotic cuisine leads to a significant increase in ethnic restaurants and sales of exotic food [11,15,16]. Currently, the number of sushi restaurants outside Japan is estimated to be between 14,000 and 18,000 [17]. It is well known that the eating habit of consuming raw or undercooked fish dishes (e.g., sushi, sashimi, ceviche, tartare) represents a considerable risk of transmitting FBZH [6,18]. Besides human and societal behaviour, environmental variables (abiotic and biotic) of aquatic environments could directly influence the biology and epidemiology of parasites. The relationships between host–parasite systems confine their occurrence to regions (e.g., endemic areas) characterized by distinct biotic and abiotic factors that align with the parasite’s life cycle. Nevertheless, substantial shifts in climatic and hydrological conditions have resulted in the alteration of the geographical distribution of parasites [19,20]. Moreover, fluctuations and changes in abiotic/biotic factors may profoundly affect parasite–host systems, resulting in alterations to host biology (species performance, population dynamics, and distribution) and parasite prevalence [21]. In light of these considerations, it is clear that the distribution of parasites is the result of an intricately interconnected relationship between hosts, parasites, and the environment [22]. Despite recent epidemiological updates and increased attention from the scientific community, food-borne parasites remain neglected in comparison to other pathogens. There are various reasons behind the lack of attention given to this group of pathogens. In fact, several parasitic diseases that induce a chronic progression are considered less relevant or urgent. Furthermore, the long incubation periods of these parasitic infections may reduce the perception of their impact on health and lead to under-diagnosis [23]. In addition, notifying public health authorities is optional for most parasitic diseases. Official reports do not reflect the real prevalence or incidence of the diseases, which result in them being under-reported; therefore, their global impact on public health is unknown, mainly due to limited data. The emergence and recrudescence of parasitic diseases make it crucial to develop an understanding of parasites’ roles in fish populations and to define their impact on fish and human health. In this context, the European Food Safety Authority (EFSA) produced a document titled “Scientific opinion on risk assessment of parasites in fishery products”, which emphasized the need to outline the risk for consumers based on epidemiological studies of parasites in fishery products [24]. Freshwater fish are pivotal to the transmission of zoonotic helminths, representing a significant public health concern, especially in regions adjacent to water basins, such as the great sub-alpine lakes like Iseo, Como, Garda, and Maggiore. Here, the consumption of freshwater fish products is historically embedded in the culinary culture and draws numerous tourists. Italy, with its extensive river networks, lakes, and diverse freshwater ecosystems, represents an ideal environment for the occurrence and transmission of freshwater fish-borne zoonotic parasites. The aim of this systematic review was to collect evidence available in the scientific literature in order to gather information on the occurrence of freshwater FBZH in Italy and neighbouring countries. The retrieved data could be useful for public health authorities in identifying the freshwater fish species and investigating the aquatic environments and epidemiological determinants of this sanitary issue.

## 2. Materials and Methods

### 2.1. Review Question, Eligibility Criteria, Information Sources, and Search Strategies

The review question was: “What is the occurrence of freshwater FBZH in Italy and neighbouring countries?” Key elements were identified as: (i) population: fish, and (ii) outcome: freshwater FBZH (see Appendix A for the complete list of keywords). We considered all studies published in peer-reviewed journals in PubMed, Embase, and Web of Science Core Collection, without time limits. The first date searched was 7 March 2023, and an update of the search was conducted on 1 August 2023. The following criteria were used to select eligible studies: (i) English language and full-text availability; (ii) the study had to report data on freshwater FBZH detection; (iii) the study had to search freshwater FBZH as consequence of natural contamination (observational study); (iv) the study had to deal with farmed/captive or wild freshwater fish (food and/or feed and/or restocking and/or research purpose); (v) the study had to involve Italy or neighbouring countries (i.e., Austria, Croatia, France, Germany, Liechtenstein, Slovenia, and Switzerland).

### 2.2. Selection and Data Collection Processes

The screening process was carried out by three reviewers (VM, EL, MB), who categorized all studies obtained via the initial literature search based on title and abstract. In the case of a poorly explicative abstract or in the case of doubt about the available data, the study was included and evaluated at full-text level. Each record underwent a dual evaluation process, where two separate reviewers independently assessed it based on the preselected eligibility criteria (EL and MB). A third reviewer solved any conflicts (VM). After full-text retrieval, one reviewer (VM) extracted data from the included studies. Data were extracted from text, tables, or figures, and were entered into pre-defined tabular forms. Extracted data were controlled by another reviewer (EL), crosschecking the extracted data with the original data in the studies. Additionally, thanks to the in-depth knowledge of the subject of the authors, four additional papers (named “other research” in the Appendix A) were included in the selection.

### 2.3. Data Items

We defined “study” as an investigation reporting data for freshwater FBZH in fish samples in Italy and neighbouring countries. General data related to the included studies were listed in tables reporting the following information: (i) fish species involved (scientific name, common name, number of total specimens sampled, and number of specimens positive for freshwater FBZH); (ii) infection indices (prevalence, mean intensity, and mean abundance); (iii) freshwater FBZH details (scientific name, morphological and/or molecular identification); (iv) country where the study was carried out; (v) details of the water collection; (vi) year and month of sampling.

### 2.4. Synthesis Methods

The data synthesis was presented in tables reporting, for each species of freshwater FBZH, details of its lifecycle and its occurrence in the study area: (i) country; (ii) type of water basin it was detected in; (iii) fish species; (iv) number of positive fish among the total analysed; (v) parasitological indexes; (vi) relative references.

### 2.5. Quality Assessment

Quality assessment was conducted taking into account relevant aspects for freshwater FBZH identification and sample characteristics to be considered in an observational study: (i) the presence of at least one of the three parasitological indexes (i.e., prevalence, mean intensity, mean abundance); (ii) parasite species morphological identification using published identification keys; (iii) molecular identification of the parasite species; (iv) geopositioning of the sampling site at least at municipality level, namely Local Administrative Unit (LAU) level 2, that is the lower level assigned for the Nomenclature of Territorial Units (NUTs) for statistics in the European Union; (v) the months in which the sampling activity occurred. The quality assessment of the included studies was carried out by one reviewer (MB) and verified by a second reviewer (EL). Each positive answer resulted in a point score; at the end of the assessment, the score ranged from 0 to 5.

## 3. Results

### 3.1. Study Selection

In total, 24 papers investigating the occurrence of freshwater FBZH in the considered countries were included after the screening process (Figure 1).

**Figure 1 animals-13-03793-f001:**
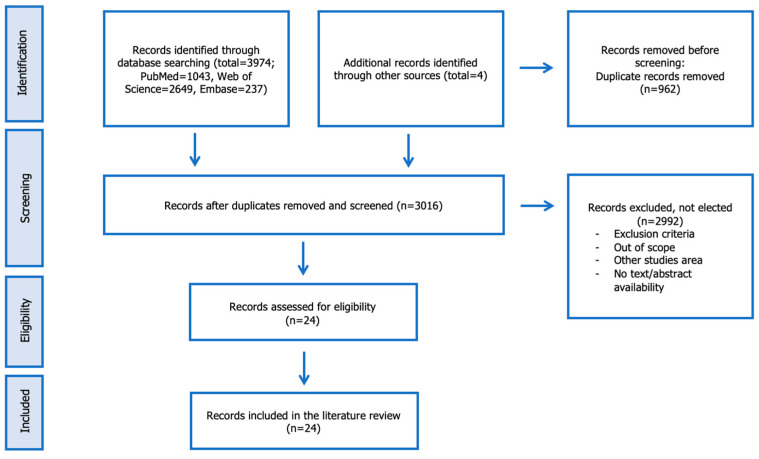
The PRISMA flow chart presents the results of the literature searches and the screening process.

Reviewing the 24 included papers, seven parasite species (i.e., *C. complanatum*, *C. rudolphii*, *D. latus*, *E. excisus*, *E. mergorum*, *O. felineus*, and *P. truncatum*) were identified in the countries of interest and described in 32 fish species from several water basins, most of them located in Italy (91%, 195/214 sites). The occurrence of FBZHs was registered only in wild fish species. This is in agreement with the literature data on other freshwater [25] and marine [26,27,28] species, stating that the risk related to the consumption of aquaculture products from European facilities is null or negligible. The absence of zoonotic parasites in aquaculture products, which was more extensively investigated for marine FBZHs, is mainly related to the use of commercial pellets as feed, which avoids the predation of infected intermediate hosts by farmed fish [29,30]. Indeed, the environment in which fish are farmed or caught directly influences the species and number of parasites in fish products [31]. An exhaustive overview that provides relevant information on the management of the top six parasites (*C. complanatum*, *C. rudolphii*, *D. latus*, *E. excisus*, *O. felineus*, and *P. truncatum*) is reported below. Given the limited presence of reports of *O. felineus* and *P. truncatum* in Italy, the authors decided to include Caffara et al. [32] despite the main aim of the paper indeed being different from the other selected papers (development of a Multiplex PCR method). In Table 1, Table 2, Table 3 and Table 4, the relevant information retrieved from the elected article is summarized for each parasite species, with the exception of *C. rudolphii* and *P. truncatum*, detected only by Mattiucci et al. [33] and Caffara et al. [32], respectively, and reported in their specific section. *Eustrongylides mergorum*, reported only by Jakob et al. [34], has not been considered in this work because it is not listed as a valid species. In fact, the valid species comprise *E. excisus*, *E. ignotus*, and *E. tubifex*. In addition, the biological cycle of the FBZHs considered in this review has been graphically represented (see Figure 2, Figure 3, Figure 4 and Figure 5). To briefly summarize all the geographical location and the type of water body considered of the retrieved papers, Figure 6 and Table 5 have been added after the FBZHs description chapters. Complete information on fish species, parasite species, sampling site, and sampling period are available in the Appendix A.

### 3.2. Clinostomum complanatum (Rudolphi, 1814, Digenea: Clinostomidae): Clinostomiasis, Life cycle and Its Occurrence in the Study Area

*Clinostomum complanatum* is a digenean trematode belonging to the Clinostomidae family. The *Clinostomum* genus comprises 16 valid species. Nonetheless, due to the significant morphological diversity within the genus, the taxonomy remains convoluted, having undergone several revisions [35,36,37,38]. In humans, *C. complanatum* is the causative agent of Halzoun syndrome, a laryngeal pharyngitis provoked by the consumption of raw fish products containing vital parasites. The metacercariae excyst within the stomach and subsequently migrate to the throat’s mucosa, where they firmly attach [39,40,41,42,43]. It has been hypothesized that *C. complanatum* feeds on blood extracted from the host’s throat, leading to mucosal haemorrhage due to the adhesion-induced injury [40]. The therapeutic approach involves surgical endoscopic remotion of the parasite from the adhesion site. Notably, treatment is often challenging due to the trematode’s rapid mobility and strong adhesion to host tissues [41]. Human infection is rare, and it has not been reported in European countries [44]. However, since the first documented human case in Japan in the early 1900s [42], numerous cases have been reported in Asian countries, with a particular concentration in Korea [39,45,46,47]. Similarly to other species in this genus, *C. complanatum* has a heteroxenous life cycle involving two intermediate hosts and one definitive host [48] (Figure 2). In the definitive host, a fish-eating bird (e.g., *Ardeids*, the adult stage parasitises the buccal cavity and releases eggs into the aquatic environment through faeces [49,50]. These eggs then hatch into miracidia within a freshwater gastropod mollusc, the first intermediate host. Subsequent development involves sporocysts, rediae, and cercariae [51,52]. The cercariae exit the mollusc, swimming in the water column to encyst within the second intermediate hosts (fish and amphibians), where they mature into metacercariae, and the infective stage for the final hosts [53,54,55]. Although *C. complanatum* is rarely documented in European countries, a recent paper reported its presence in a *Perca fluviatilis* population from Endine Lake, a small Italian lake located in the sub-alpine area [44]. Previously the parasite had been detected in *Cobitis bilineata* caught in water basins in Piedmont [56] and in two species in Emilia Romagna region [57].

**Figure 2 animals-13-03793-f002:**
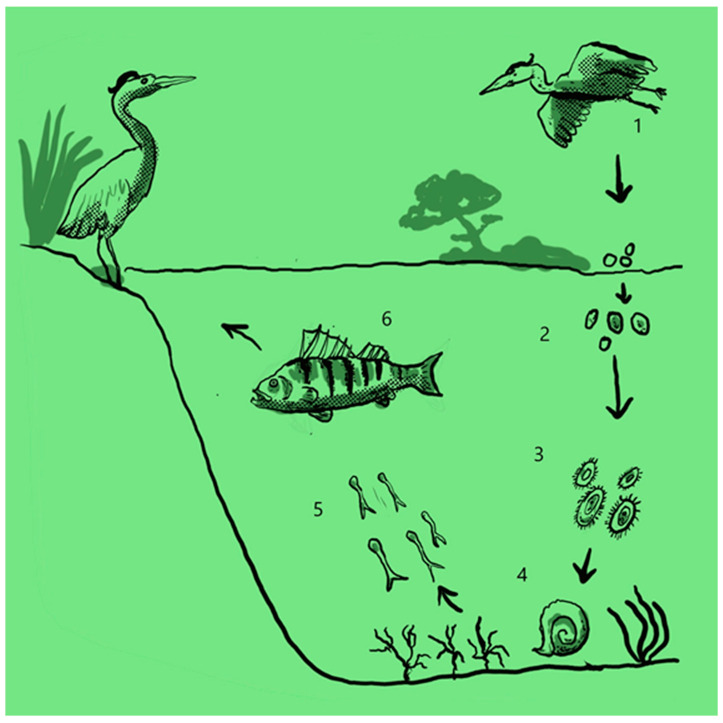
Life cycle of *C. complanatum*. Note: 1: Bird, the final host; 2: eggs, 3: Miracidia; 4: gastropod mollusc as the first intermediate host; 5: Cercariae; 6: fish as the second intermediate host.

**Table 1 animals-13-03793-t001:** Sampling location, fish host, and the parasitological indexes of *C. complanatum*.

Country	Water Basin	Year	Fish Species	P/N	Parasitological Indexes			References
					P%	MI	MA	
Italy	Sillaro, Soligo, and Santerno rivers	N/A	*B. barbus*	N/A	N/A	N/A	N/A	[57]
		N/A	*S. cephalus*	N/A	N/A	N/A	N/A	
Italy	Water basins in Piedmont	2014	*C. bilineata*	12/30	40	2	0.8	[56]
Italy	Lake Endine	2019	*P. fluviatilis*	21/112	18.75	1.33	0.25	[44]

Note: P/N: N of positive specimens/N specimens; P%: prevalence; MI: mean intensity; MA: mean abundance; N/A: not available.

### 3.3. Contracaecum rudolphii (Hartwich, 1964, Nematoda: Anisakidae): Anisakidosis, Life Cycle, and Its Occurrence in the Study Area

Parasites of the genus *Contracaecum* are ascaridoid nematodes belonging to the family Anisakidae. With more than 100 species, *Contracaecum* is the most numerous genus in the Anisakidae family [58]. The taxonomic status of *C. rudolphii* comprises a complex of sibling species as revealed by molecular studies [59,60,61]. In particular, multilocus emanzyme electrophoresis was used to understand the extent of genetic variation among specimens from different geographical regions. This approach identified the presence of two closely related species within the *C. rudolphii* sensu lato group, respectively *C. rudolphii* A and *C. rudolphii* B [61] in Europe. Hosts play a significant role in improving the genetic diversity of parasites by facilitating gene exchange among different parasite populations, across various geographical areas [62]. Human anisakidosis is a parasitic disease caused by the ingestion of raw or undercooked fish products containing vital third-stage larvae of Anisakidae nematodes, including the *Contracaecum* species [63,64,65,66]. However, human infections caused by *Contracaecum* larvae have been reported much less frequently compared to infections caused by *Anisakis* larvae [67]. The disease caused by *Contracaecum* larvae leads to a severe and painful condition in humans, characterized by gastrointestinal pain, vomiting, and diarrhoea [66]. The massive presence of *Contracaecum* larvae in fish products can lead to market rejection and impose economic repercussions [68]. The life cycle of *C. rudolphii* is complex and remains partially understood. This parasite exhibits a broad host range, with 24 genera of fish-eating birds, including pelicans, herons, and mergansers, serving as definitive hosts [69]. In Europe, primary definitive hosts are cormorants (*Phalacrocorax carbo*) and sea ducks (*Mergus merganser* and *Mergus serrator*) [70,71]. Lesions in the avian host include fibrinous haemorrhagic exudates over the intestinal serosa. There are reports of mortality episodes in American Double-crested Cormorants (*Nannopterum auritum*) due to severe infections of hundreds of individuals of ascaridoids (*C. multipapillatum*, *C. microcephalum*, and *C. rudolphii* sensu lato) [72]. The adult stage resides in the proventriculus and intestines of birds, where they produce eggs after fertilization of the female by the male. These eggs lack a chitinous membrane, making them sensitive to physical and chemical factors and mechanical damage [73]. Avian definitive hosts excrete these eggs into aquatic environments through their faeces, giving rise to the second larval stage, a free-living larva [74]. When larvae are ingested by planktonic or benthic invertebrates (the first intermediate host), they grow within the host’s body without moulting [75]. Fish become infected when they consume these infected invertebrates containing the larvae L2. These larvae reach the abdominal cavity, grow and later moult to L3. The third-stage larvae then encyst themselves within the fish’s intestinal wall or migrate to other internal organs and encapsulate there. When infected fish are consumed by a final host, L3 larvae undergo two moults to reach the adult stage [76]. Among the retrieved papers, this parasite was detected only by Mattiucci et al. [33] in two Italian water bodies, Transimeno Lake and the River Marta, located in the Lazio region. The sibling species *C. rudolphii* A larvae was detected in coinfection with *C. rudolphii* B in fish from Trasimeno Lake (*Carassius carassius* and *Atherina boyeri*), while only *C. rudolphii* B was reported from freshwater fish species from the River Marta (*Leuciscus cephalus*, *Barbus barbus*) [33].

### 3.4. Dibothriocephalus latus (Linnaeus, 1758, Cestoda: Diphyllobothriidea; syn. Diphyllobothrium latum): Dibothriocephalosis Life Cycle, and Its Occurrence in the Study Area

Diphyllobothriosis is a fish-borne zoonosis caused by fish tapeworms of different genera belonging to the family Diphyllobothriidae. Based on newly published data, approximately 20 million individuals are believed to be affected on a global scale. Nevertheless, the exact prevalence of *D. latus* across the world remains uncertain [77]. One of the main causative agents of human diphyllobothriasis is *D. latus*. Generally, human infections are reported as asymptomatic or present mild clinical pictures and variable symptomatology [78]. Among 25% of the cases may manifest diarrhoea, abdominal pain, fatigue, headache, constipation, intestinal obstruction, and, sporadically, a non-lethal form of pernicious anaemia [79]. Haematological alterations result from the fast intestinal absorption of vitamin B12 caused by these parasites, which are attached to intestinal mucosa [77]. The diagnosis of diphyllobothriasis is based on coprological examination, aimed at detecting parasites’ eggs and proglottids. Clinical treatment involves anthelminthic drugs (praziquantel or niclosamide), which are commonly quite effective and make it possible for parasites to be expelled from infected patients [80]. The most effective way to prevent diphyllobothriasis is avoiding the consumption of raw or improperly treated fish products. At the beginning of the 20th century, Italy classified diphyllobothriasis as a notifiable disease due to the high prevalence of human cases [81]. Although a few isolated cases were noted later on, diphyllobothriasis was generally considered a minor public health concern [82,83,84]. However, in the last 20 years, a rising number of human diphyllobothriasis cases was observed in Italy, France, and Switzerland [85,86,87]. In fact, the endemic area of this parasite mainly concerns the great alpine lakes of these three countries [86,88,89,90,91,92,93].

The *D. latus* life cycle involves two intermediate hosts, a crustacean copepod (Cyclops) and a predatory freshwater fish (*P. fluviatilis*), and one definitive host, a fish-eating mammal (including humans) [87] (Figure 3). The parasites can also be transmitted to other paratenic hosts such as burbot (*Lota lota*), and pike (*Esox* sp.), and finally consumed by the definitive host [94]. The adult stage inhabits the intestine of the definitive host and produces eggs that are released through faeces. The resistance of these eggs to external factors is quite low and, for hatching, they must quickly reach a suitable water body [95]. The coracidium (first larval stage) swims actively and attracts predation by crustacean copepods, where it develops into a procercoid larva [96]. When the second intermediate host, a predatory fish, consumes an infected copepod, it contracts the parasite, which migrates from the intestine to the skeletal muscle, where it develops into the plerocercoid larva [87]. In *P. fluviatilis* the localization of plerocercoid larvae in the visceral cavity occurs rarely, while in pike and burbot mainly concerns the visceral cavity [89,94]. Humans acquire the infection by consuming raw or undercooked fish products containing viable plerocercoid larvae. The plerocercoid larvae migrate after ingestion to the intestine of the definitive host and develop into the mature stage. The reproductive potential of *D. latus* is extremely high (1 million eggs per day), so even sporadic human cases can give rise to a high prevalence in a fish population [97]. Fecal contamination of water bodies plays a pivotal role in maintaining the *D. latus* life cycle and the persistence of diphyllobothriasis in endemic areas [86,87,95]. The selected articles show a significant prevalence of *D. latus* in the fish population of sub-alpine lakes, undoubtedly reconfirming this region as an endemic area. In particular, concerning *P. fluviatilis*, the highest prevalence values were detected in Lake Como (46.5%) [98], followed by Lake Biel (37.5%) [94] and Lake Iseo (22.8%) [94]. The analysis of historical data shows wide variations in prevalence among studies conducted on the same lake (see Appendix A). For example, in perch sampled from Lake Iseo, the prevalence ranged from 7.6% [89] to 22.8% [94]. Regarding Lake Como [98], the provided prevalence values range from 18.7% to 46.5%, depending on the sampling areas in the lake (Western, Eastern, North, and Centre); however, the sampling points were not georeferenced. It is clear that the total prevalence data of a lake are not representative and could be misleading for evaluating parasite occurrence. Accordingly, the latest papers on *D. latus* in Italy reveal new information on spatial distribution and show significant differences in prevalence values among the sampled areas of Lake Iseo [87,90]. In particular, Menconi et al. [87] provided a spatial analysis of *D. latus* occurrence in *P. fluviatilis*. This study georeferenced sampling points and associated the prevalence of *D. latus* with water *E. coli* load, demonstrating a significant relationship between these variables.

**Figure 3 animals-13-03793-f003:**
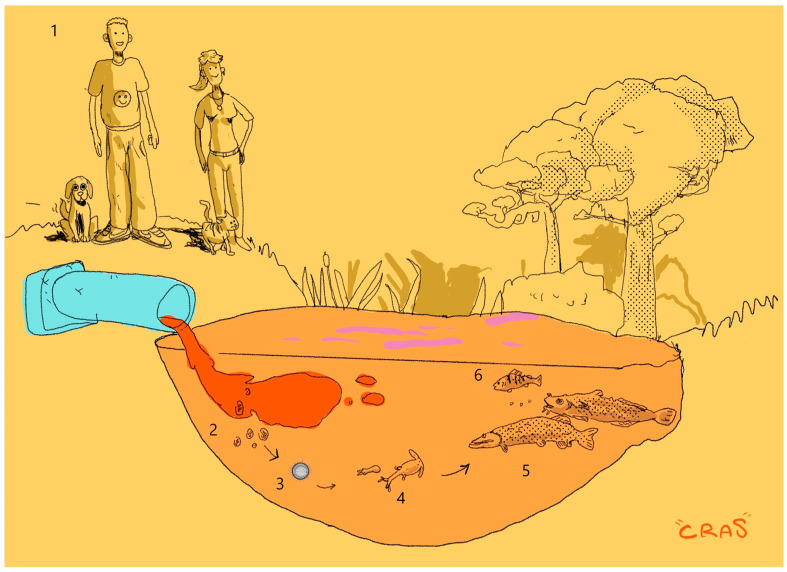
Life cycle of *D. latus.* Note: 1: definitive host; 2: eggs; 3: coracidia; 4: copepods crustacean—first intermediate host; 5: pike and burbot—paratenic host; 6: European perch—second intermediate host.

**Table 2 animals-13-03793-t002:** Sampling location, fish host, and the parasitological indexes of *Dibothriocephalus latus*.

Country	Water Basin	Year	Fish Species	P/N	Parasitological Indexes			References
					P%	MI	MA	
Italy/Switzerland	Lake Maggiore	N/A	*P. fluviatilis*	24/309	7.8	1	0.72	[99]
Italy	Lake Orta		*P. fluviatilis*	5/15	33.3	1	0.33	
Italy	Lake Como	2005/2007	*P. fluviatilis*	183/609	30	1.25	N/A	[98]
Italy/Switzerland	Lake Maggiore	2005–2008	*P. fluviatilis*	123/880	14	N/A	N/A	[91]
Switzerland	Lake Geneva	2006–2008		27/532	5.1	N/A	N/A	
Italy	Lake Iseo	2013–2014	*P. fluviatilis*	35/458	7.6	1.29	0.1	[89]
			*L. lota*	1/26	3.80	0.12	3	
			*E. Lucius*	5/7	71.4	16.4	11.71	
	Lake Como	2013–2014	*P. fluviatilis*	108/426	25.4	1.24	0.31	
			*L. lota*	2/55	3.6	1.5	0.05	
			*E. lucius*	16/19	84.2	28.25	23.79	
	Lake Maggiore	2013–2014	*P. fluviatilis*	42/635	6.6	1.05	0.07	
			*E. lucius*	1/1	100	1	1	
Italy	Lake Iseo	2017	*P. fluviatilis*	19/148	12.8	N/A	N/A	[94]
		2018		53/232	22.8	N/A	N/A	
	Lake Como	2017	*P. fluviatilis*	7/46	15.2	N/A	N/A	
Italy/Switzerland	Lake Maggiore	2017	*P. fluviatilis*	8/48	16.7	N/A	N/A	
Switzerland	Lake Biel	2018	*P. fluviatilis*	3/8	37.5	N/A	N/A	
Switzerland	Lake Neuchâtel	2018	*P. fluviatilis*	1/50	2	N/A	N/A	
Switzerland/France	Lake Geneva	2018	*P. fluviatilis*	10/156	6.4	N/A	N/A	
Italy	Lake Iseo	2019	*P. fluviatilis*	39/598	6.5	1.07	0.07	[90]
Italy	Lake Iseo	2020	*P. fluviatilis*	45/550	8.1	N/A	N/A	[87]

Note: P/N: N of positive specimens/N specimens; P%: prevalence; MI: mean intensity; MA: mean abundance; N/A: not available.

### 3.5. Eustrongylides excisus (Jagerskiold, 1909, Nematoda: Dioctophymatidae): Eustrongylidiasis, Life Cycle, and Its Occurrence in the Study Area

*Eustrongylides* spp. are nematodes belonging to the family of Dioctophymatidae [100]. The taxonomic status of the genus *Eustrongylides* has recently been subjected to several revisions, and molecular taxonomy leads to considering the following three species valid: *E. tubifex*, *E. excisus*, and *E. ignotus*. *Eustrongylides* nematodes are responsible for rare human infections; however, they could potentially impact public health [14]. According to current knowledge, five human cases have been reported, three in the USA [101,102,103] and two in Africa [104]. Human eustrongylidiasis is related to the consumption of infected raw fish meat. The infection is characterized by acute pain that occurs 24 h after the ingestion of the contaminated fish product, due to the larvae penetration into the gut wall. Medical treatment of the patients requires surgical removal of the encysted larvae [104,105]. *Eustrongylides* nematodes have a complex life cycle, including five developmental stages, and involving two intermediate hosts and one definitive host. The focus of infection is associated with freshwater environments [106] (Figure 4). The adult stage parasitises the mucosa of the oesophagus, proventriculus, and/or intestine of piscivorous birds (Ciconiformes and Phalacrocoracidae). The final hosts release eggs through their stool and, if ingested by aquatic oligochaetes (*Tubifex*, *Lumbricus*, and *Limnodrilus*), they develop into the second and third larval stages. Eutrophication of the water basin provides a high concentration of nutrients and facilitates the growth of oligochaete populations and could, therefore, play a pivotal role in the occurrence of *Eustrongylides* [107]. Several fish species (17 taxonomic orders) are known as second intermediate hosts [106,107,108,109,110]. Predatory fish species, such as perch and largemouth bass, have a greater potential to become highly infected by consuming infected fish, which amplifies the intensity of parasite infection [109]. Moreover, large predatory fishes, amphibians (Bufonidae), and reptiles (*Natrix tessellata*) may act as paratenic hosts and could infect final hosts [111,112]. Nematodes ascribable to the genus *Eustrongylides* are now widespread and reported on all continents [113]. The distribution of the dominating *Eustrongylides* species varies among countries; *E. ignotus* and *E. tubifex* are generally reported in North America, while *E. excisus* is common in Europe and the Middle East [100]. Bibliographical analysis indicates that *E. excisus* had never been reported in European countries prior to 2009 [34]; however, no further reports were made until 2015 [114]. After the report by Dezfuli et al. [114], this parasite has gained attention and numerous studies have been published (Table 3). Recent papers show an expanding geographical distribution of *Eustrongylides* nematodes in Italy, potentially linked to the growing cormorant population concentrated around lakes and other biotic and abiotic factors [109,115]. The expansion of this parasite is a multifactorial phenomenon, and understanding the interactions among key factors in the *Eustrongylides* life cycle is essential to comprehending its epidemiology. To trigger an epizootic event, multiple elements must occur: eutrophic wetlands supporting dense fish populations, infected birds attracted to fish, and nutrient-polluted sites providing suitable habitats for oligochaetes [69,116]. It is important to mention that, except for Germany [34], available data in this area of study exclusively concerns Italian lakes. *Eustrongylides* occurrence is widely registered in comestible and commercially relevant fish species such as perch (*P. fluviatilis*), largemouth bass (*Micropterus salmoides*), and sand smelt (*A. boyeri*) [109,114,117,118,119,120]. Among these fish species, the highest prevalence value was recorded in perch from Lake Trasimeno (67.99%) [120]. The values previously found by Dezfuli et al. (6%) [114] and Branciari et al. (6.84%) [117] were instead significantly lower. Following, the highest prevalence values are attributable to *M. salmoides* (30%) fished in Lake Garda [109] and *Lepomis gibbosus* (18.3%) from Lake San Michele [118]. Bibliographic data reports the occurrence of *Eustrongylides* nematodes in new fish hosts including eel (*Anguilla anguilla*), black bullhead catfish (*Ameiurus melas*), tench (*Tinca tinca*), carp (*Cyprinus carpio*), wels catfish (*Silurus glanis*), and small thin lip grey mullet (*Chelon ramada*) [34,116,120]. The expansion of the range of fish species susceptible to *E. excisus* highlights the importance of ongoing parasitological research in fish species of commercial significance, especially in water basin where professional fishing is highly active.

**Figure 4 animals-13-03793-f004:**
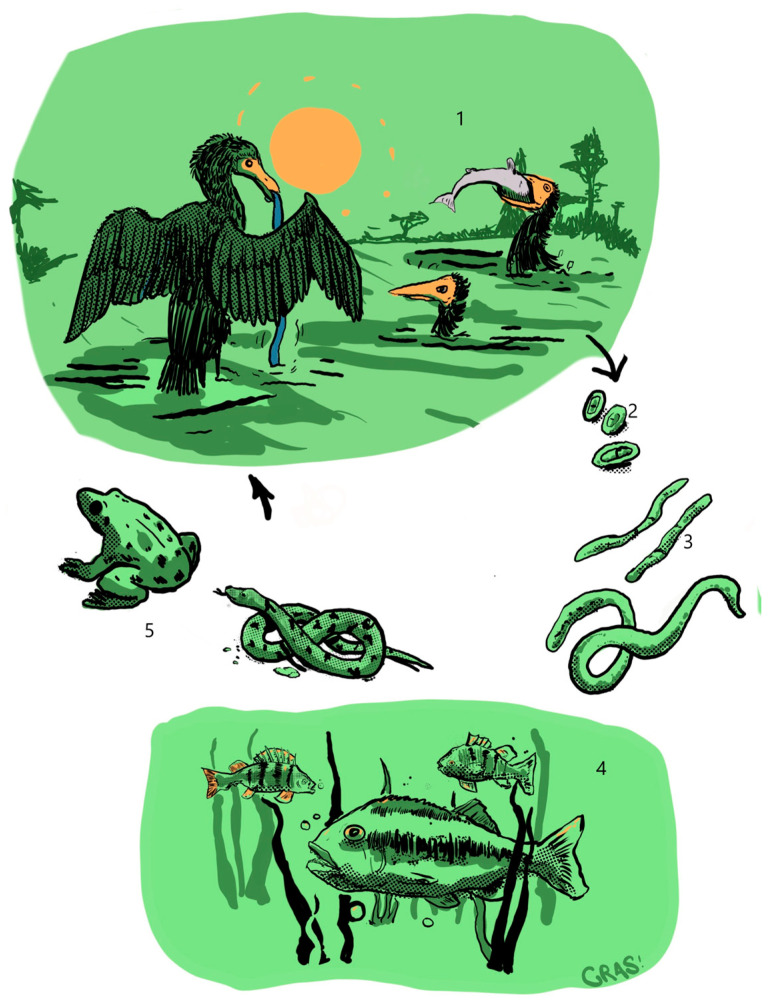
Life cycle of *E. excisus*. Note: 1: bird host; 2: eggs; 3: oligochaete as first intermediate host; 4: fish as second intermediate host; 5: amphibians and reptiles may act as paratenic hosts.

**Table 3 animals-13-03793-t003:** Sampling location, fish host, and the parasitological indexes of *Eustrongylides excisus*.

Country	Water Basin	Year of Sampling	Fish Species	P/N	Parasitological Indexes			References
					P%	MI	MA	
Germany	Lake Plon	2006	*A. anguilla*	16/30	53.3	2.1	N/A	[34]
Italy	Lake Trasimeno	2014	*P. fluviatilis*	31/510	6	N/A	N/A	[114]
Italy	Lake Trasimeno	2015	*P. fluviatilis*	105/1536	6.84	1.30	N/A	[117]
			*M. salmoides*	29/1536	1.89	1	N/A	
			*A. boyeri*	1/768	0.13	1	N/A	
Italy	Lake Trasimeno	N/A	*P. fluviatilis*	5/5	100	N/A	N/A	[121]
			*A. boyeri*	2/2				
Italy	Lake Garda	2020	*P. fluviatilis*	3/212	14.41	1	0.014	[109]
			*M. salmoides*	6/20	30	1.33	0.4	
			*L. gibbosus*	4/129	3.10	1	0.023	
Italy	Lake Massacciuccoli	2019	*A. boyeri*	75/3317	2.3	1	0.2	[119]
Italy	Lake Trasimeno	2016	*P. fluviatilis*	4696/111,235	4.22	1	0.04	[120]
		2017		961/23,318	4.12	1	0.04	
		2018		12,288/89,094	13.68	1.26	0.17	
		2019		19,016/56,962	33.38	3.42	1.14	
		2020		71,522/128,979	55.45	7.05	3.91	
		2021		27,890/41,021	67.99	8.28	6	
		2016	*M. salmoides*	143/30,334	0.47	1	0.005	
		2017		148/81,964	0.18	1	0.002	
		2018		162/40,296	0.40	1	0.4	
		2019		485/77,494	0.63	1	0.01	
		2020		72/23,498	0.30	1	0.003	
		2021		5/1929	0.26	1	0.003	
		2016	*T. tinca*	5551	N/A	N/A	N/A	
		2017		5812	N/A	N/A	N/A	
		2018		1/2801	0.036	1	0.0004	
		2019		1/1801	0.06	1	0.0006	
		2020		3/3052	0.10	1	0.001	
		2021		1/792	0.13	1	0.0013	
		2016	*A* *. melas*	680	N/A	N/A	N/A	
		2017		1079	N/A	N/A	N/A	
		2018		1/1386	0.07	1	0.0007	
		2019		1/2623	0.04	1	0.0004	
		2020		1/2820	0.04	1	0.0004	
		2021		3/110	2.72	1	0.0273	
		2016	*C. carpio*	1867	N/A	N/A	N/A	
		2017		2720	N/A	N/A	N/A	
		2018		1/4062	0.02	1	0.0002	
		2019		1/4931	0.02	1	0.0002	
		2020		1/5719	0.02	1	0.0002	
		2021		1/5777	0.02	1	0.0002	
Italy	Lake Annone	2019	*P. fluviatilis*	11/114	9.66	N/A	N/A	[122]
Italy	Lake Massaciuccoli	2021–2022	*A. melas*	22/77	28.6	1.7	0.49	[116]
			*S. glanis*	42/56	75	5.7	4.3	
			*C. ramada*	258	N/A	N/A	0.07	
			*L. gibbosus*	1/23	4.3	1	0.04	
			*M. salmoides*	1/4	25	1	0.25	
			*A. boyeri*	180/3500	5.1	1	0.05	

Note: P/N: N of positive specimens/N specimens; P%: prevalence; MI: mean intensity; MA: mean abundance; N/A: not available.

### 3.6. Opisthorchis felineus (Rivolta, 1884, Trematoda: Opisthorchiidae): Opisthorchiasis, Lifecycle, and Its Occurrence in the Study Area

The family of Opisthorchiidae is a large group of trematodes that comprise 33 genera [123,124]. *Opisthorchis felineus* (Rivolta, 1884), *Opisthorchis viverrini* (Poirier, 1886), and *Clonorchis sinensis* (Cobbold, 1875) compose the triad of the primary agents of opisthorchiasis, affecting millions of people worldwide, especially in Asiatic countries [125]. *Opisthorchis viverrini* and *C. sinensis* are the most significant species reported, respectively, in Southeast Asia (Cambodia, Laos, Myanmar, Thailand, and Southern Vietnam) and East/Southeast Asia (China, Eastern Russia, Northern Vietnam, and South Korea) [6]. The main endemic area of *O. felineus* is associated with the Ob-Irtisch River basin (Russia) [124]. However, since 2003, human cases have been increasingly reported in Italy and other EU countries, with more than 200 human infections registered [126,127,128]. Opisthorchiasis is related to the consumption of raw/undercooked contaminated fish products [129,130]. Clinical manifestations depend on the number of ingested metacercariae. A low level of parasite infestation may cause mild or asymptomatic infections and result in a delayed or missed diagnosis, which could lead to chronic disease patterns. *Opistorchis felineus*’ chronic infections may induce severe liver disorders associated with the hepatobiliary system as cholecystitis, cholelithiasis, cholangitis, and periductal fibrosis [131,132]. Since *Opisthorchis* sp. cannot multiply in humans, high-intensity infections result from repeated exposures to contaminated raw fish preparations [133]. *Opisthorchis viverrini* and *C. sinensis* are considered biological carcinogens and classified as significant risk factors for cholangiocarcinoma, while *O. felineus* is not officially considered carcinogenic to humans [134]. Similar to other species within the Opisthorchiidae family, *O. felineus* has a complex life cycle linked to the freshwater environment, with two intermediate hosts and a definitive host. In particular, the life cycle involves a gastropod snail (*Bithynia* sp.) and a fish (Cyprinid) as the first and second intermediate hosts. Domestic cats and other fish-eating mammals (e.g., foxes), including humans, act as definitive hosts (Figure 5). The adult stage lives in biliary and pancreatic ducts, where it deposits eggs, which are released through faeces [135]. The eggs, when ingested by appropriate snail species, release the miracidia. Inside the snail, miracidia undergo three developmental stages, sporocysts, rediae, and cercariae [129]. Free-swimming cercariae leave the mollusc and, once they reach a fish host, they penetrate and encyst into the muscular tissue and develop into metacercariae, the infective stage for the final host [127,136]. Under favourable conditions (e.g., water temperature and light), the life cycle is completed within four or five months. Our data indicates that the occurrence of *O. felineus* in fish populations in Europe is exclusively reported in Italy, except for a record in Germany. In the temperate areas of Europe and Asia, *O. felineus* is described in at least 23 species belonging to the Cyprinidae family [124]. In the study area, *T. tinca* (Italy) and *Rutilus rutilus* and *A. alburnous* (Germany) are identified as suitable fish hosts [129,130].

**Table 4 animals-13-03793-t004:** Sampling location, fish host and the parasitological indexes of *Opisthorchis felineus*.

Country	Water Basin	Year	Fish Species	P/N	Parasitological Indexes			References
					P%	MI	MA	
Germany	Finow canal	1995	*R. rutilus*	38/50	76	N/A	N/A	[129]
		1995	*S. erythrophthalmus*	8/50	16	N/A	N/A	
		1995	*A. alburnus*	37/50	74	N/A	N/A	
		1995	*A. brama*	18/50	36	N/A	N/A	
		1995	*B. ballerus*	1/11	14	N/A	N/A	
		1995	*B. bjoerkna*	2/16	16	N/A	N/A	
Italy	Lake Bolsena	2007–2008	*T. tinca*	33/44	75	N/A	N/A	[130]
	Bracciano	2007–2008		83/87	95.4	N/A	N/A	
Italy	Lake Bolsena	N/A	*T. tinca*	N/A	N/A	N/A	N/A	[32]

Note: P/N: N of positive specimens/N specimens; P%: prevalence; MI: mean intensity; MA: mean abundance; N/A: not available.

**Figure 5 animals-13-03793-f005:**
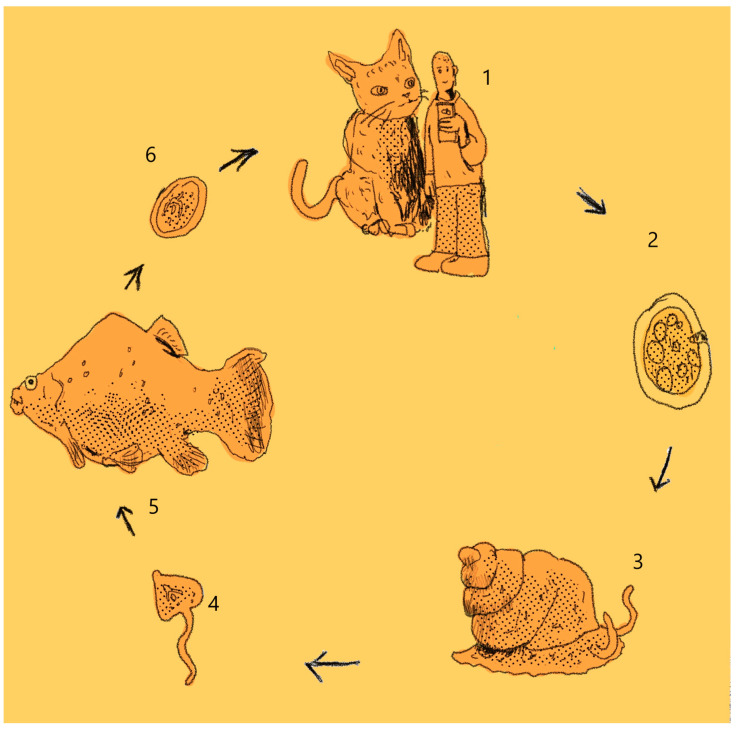
Life cycle of *O. felineus*. Note: 1: definitive host; 2: eggs; 3: gastropod mollusc as first intermediate host; 4: Cercariae; 5: fish as second intermediate host; 6: Metacercariae.

### 3.7. Pseudamphistomum truncatum (Rudolphi, 1819, Trematoda: Opisthorchiidae): Pseudamphistomiasis, Life Cycle, and Its Occurrence in the Study Area

*Pseudamphistomum truncatum* is a digenetic trematode belonging to the family of Opisthorchiidae [32]. Due to the similar symptoms caused by Opisthorchiid species, such as adenomatous hyperplasia of the biliary epithelium or biliary system obstruction, it is essential to consider the food hygiene aspect and the zoonotic potential of *P. truncatum*. Human cases of pseudamphistomiasis are rare and poorly documented, with most reports originating from the Don River basin in Southwest European Russia. Outside this region, occurrences are sporadic [137]. In general, the life cycle of *P. truncatus* closely resembles that of species belonging to the *Opisthorchis* genus [138]. *Pseudamphistomum truncatum* has an indirect life cycle involving a broad group of fish-eating mammals (e.g., otter, mink, raccoon, and dog) as definitive hosts [139]. It relies on two intermediate hosts: freshwater gastropods as the first intermediate host (*Bithynia tentaculata*) and freshwater fish (Cyprinid) as the second intermediate host [123,140]. Adult parasites lay eggs in the host’s bile duct, which are transported to the intestines along with the bile and expelled through the faeces. For further development, the eggs must reach freshwater environments and be ingested by a *Bithynia* snail. Inside the mollusc, miracidia hatch, migrate to the hepatopancreas, and develop into rediae. The occurrence of liver flukes is influenced by the presence of snail species involved in the life cycle. Therefore, these parasites are generally found in areas densely populated with specific snail species [6]. Free-swimming cercariae exit the snail and can migrate within the water column to find the fish host. Upon contacting a suitable fish host, cercariae attach, penetrate the skin, and enter the muscle tissue, maturing into metacercariae. Infection of the final host occurs when the infected fish is consumed [141]. The occurrence of *P. truncatum* is primarily documented in wild carnivore mammals across Eurasia [139,140,142]. However, except for wild tench specimens sampled at Lake Como in 2020 by Caffara et al. [32], *P. truncatum* has never been described in fish populations within the study area.

Figure 6 summarizes the water bodies positive for FBZH in the study area, each designated with a distinct ID referenced in Table 5. 

**Figure 6 animals-13-03793-f006:**
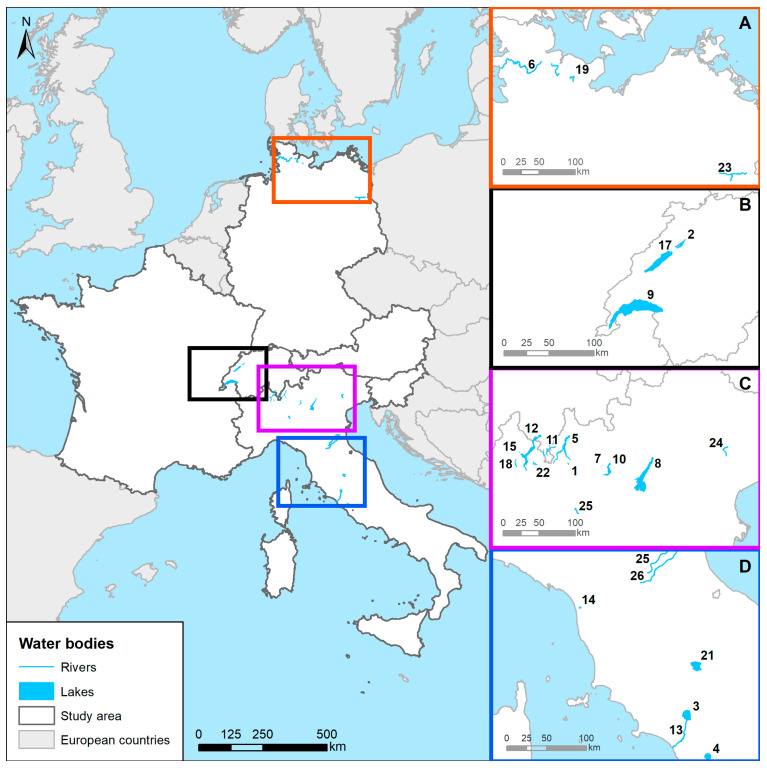
The map describes the study area and highlights the water bodies (rivers and lakes) positive for FBZH. Each spatial element is identified by an identification number (ID) is also reported in Table 5. Each macro area is identified by coloured zoom boxes presented on the right side (**A**–**D**). The map was made using ESRI™ ArcMap software version 10.8.2.

**Table 5 animals-13-03793-t005:** The list of water bodies is reported, including some basic information, and identified by an identification number as shown on the map. * ID 16, named ‘Natural fresh water basins’, was not reported on the map because the authors of the reference article did not report the spatial information.

ID	Water Body Name	Type	Country
1	Lake Annone	Lake	Italy
2	Lake Biel	Lake	Swiss
3	Lake Bolsena	Lake	Italy
4	Lake Bracciano	Lake	Italy
5	Lake Como	Lake	Italy
6	River Eider	River	Italy
7	Lake Endine	Lake	Italy
8	Lake Garda	Lake	Italy
9	Lake Geneva	Lake	Swiss
10	Lake Iseo	Lake	Italy
11	Lake Lugano	Lake	Italy
12	Lake Maggiore	Lake	Italy
13	River Marta	River	Italy
14	Lake Massaciuccoli	Lake	Italy
15	Lake Mergozzo	Lake	Italy
16	Natural fresh water basins (Piedemont region) *	Basin	Italy
17	Lake Neuchâtel	Lake	Swiss
18	Lake Orta	Lake	Italy
19	Lake Plon	Lake	Germany
20	Lake San Michele	Lake	Italy
21	Lake Trasimeno	Lake	Italy
22	Lake Varese	Lake	Italy
23	Finnow Canal	River	Germany
24	River Soligo	River	Italy
25	River Sillaro	River	Italy
26	River Santerno	River	Italy

### 3.8. Risk of Bias within Studies (Quality Evaluation)

The quality assessment of the included papers is reported in Table 6. Almost all of the papers (22 out of 24) reported at least one of the three parasitological indexes. Morphological and/or molecular identification of parasites at species level has been applied in 21 out of 24 papers. The extent (km^2^) of the water bodies examined in the included papers could be very variable. Consequently, providing precise geopositioning information (e.g., municipality, locality, or coordinates) would be a great help for advancing public health assessments and implementing preventive actions. It is noteworthy that among the retrieved papers, less than 40% (9/24) reported this level of geopositioning detail. As such, we would like to encourage authors to report precise geopositioning data of sample sites in future publications on this topic. The sampling period (months and years) is reported in 17 out of 24 papers. Among the 24 included papers, only four obtained the maximum quality assessment score (5 points) while only one study [32] found to be deficient in four quality criteria.

**Table 6 animals-13-03793-t006:** Quality assessment of included papers.

Author	Does the Paper Report at Least One of the Parasitological Indexes?	Does the Paper Use the Morphological Keys for Parasite Species Identification?	Does the Paper Identify the Parasite Species through Molecular Techniques?	Does the Paper Ensure Detailed Geopositioning of the Sampling Sites?	Does the Paper Report the Months of the Sampling Activity?	Score
Menconi et al., 2020 [118]	1	1	1	1	1	5
Menconi et al., 2020 [90]	1	1	1	1	1	5
Menconi et al., 2021 [87]	1	1	1	1	1	5
Wicht et al., 2009 [98]	1	1	1	1	1	5
Castiglione et al., 2023 [116]	1	1	1	0	1	4
Gaglio et al., 2016 [56]	1	1	1	0	1	4
Guardone et al., 2021 [119]	1	1	1	0	1	4
Gustinelli et al., 2016 [89]	1	1	1	0	1	4
Hering-Hagenbeck and Schuster, 1996 [129]	1	1	0	1	1	4
Mattiucci et al., 2020 [33]	1	1	1	0	1	4
Menconi et al., 2020 [44]	1	1	1	1	0	4
Menconi et al., 2021 [109]	1	1	1	1	0	4
Wicht et al., 2009 [91]	1	0	1	1	1	4
Branciari et al., 2016 [117]	1	1	0	0	1	3
De Liberato et al., 2011 [130]	1	0	1	0	1	3
Franceschini et al., 2022 [120]	1	0	1	0	1	3
Mazzone et al., 2019 [121]	1	1	1	0	0	3
Radacovska et al., 2020 [94]	1	0	1	0	1	3
Rusconi et al., 2022 [122]	1	1	1	0	0	3
Caffara et al., 2011 [57]	0	1	1	1	0	3
Dezfuli et al., 2015 [114]	1	0	0	0	1	2
Jakob et al., 2009 [34]	1	0	0	0	1	2
Peduzzi Boucher-Rodoni, 2001 [99]	1	0	0	1	0	2
Caffara et al., 2020 [32]	0	0	1	0	0	1

Note: 1—the paper meets the quality criteria; 0—the paper is deficient in terms of the quality criteria.

## 4. Discussion

The present study is the first systematic review on zoonotic helminths affecting the European freshwater fish population. Results show that European perch (*P. fluviatilis*) is crucial to maintain the life cycle of *D. latus* and is a suitable intermediate host for *C. complanatum* and *E. excisus* [89,90,91,94,120]. According to our results, in EU countries *P. fluviatilis* could be considered the freshwater fish species of greatest concern for FBZHs. In fact, *P. fluviatilis* has several biological features which make it an essential parasite host. This fish species is widely distributed across European freshwater basins and is commonly eaten by many piscivorous birds. Its diet comprises the parasites’ intermediate hosts, such as zooplankton, benthic invertebrates, and fish [122]. Moreover, perch is among the most important products of local fisheries of Italian lacustrine areas and is included in traditional cuisine. A popular dish derived from perch is “*Carpaccio di persico*”, which is prepared with slices of raw fish meat [77,89]. The presence of *C. complanatum* has been recorded exclusively in *P. fluviatilis* from Endine Lake, a minor basin with no professional fishing. Consequently, the risk to consumers is relatively low. However, future studies may provide new reports, showing a different epidemiological picture. The alpine region remains confirmed as the endemic area for *D. latus*, and recent epidemiological trends advocate for a geographical approach to identifying high-risk areas and enhance the efficiency of wastewater treatment plants. The recent expansion of *E. excisus* in the Italian lacustrine area is linked to several factors, which create a favourable environment for the *E. excisus*’ life cycle. The increase in the cormorant population seems to be the main factor, followed by progressive lake eutrophication and rising temperatures [107,115,143]. Given the dimensions (3–5 cm) and the reddish colour of *Eustrongylides* spp. larvae, they can negatively affect the commercial value and the marketability of infected fish products. It is important to remark that fishery products evidently contaminated with parasites are not allowed on the market for human consumption [144]. Available data on the occurrence of *O. felineus* in the study area clearly indicates which fish species and lakes are at risk. In EU countries, consumption of the Cyprinid species is marginal; however, in Italy, around lacustrine areas, tench is included in traditional cuisine. In Italy, tench is popular and is included in the culinary tradition of the areas surrounding both the large subalpine lakes (Iseo, Garda, Como, and Maggiore) as well as the lakes of central Italy (Bolsena, Bracciano, and Trasimeno). In recent times, in Northern Italy and other EU countries, there has been a growing demand for tench fish from Lakes Bolsena and Bracciano, driven by relatively local lakeside markets [127]. Considering the pathogenicity of *O. felineus* for humans, further studies are needed to define the occurrence of this parasite in European fish fauna. This study’s data on *Pseudamphistomum* and *C. rudolophii* reveals only one reported case for each, indicating a low risk for consumers. However, in light of the recent epidemiological trend observed for *E. excisus*, we should not dismiss the possibility of potential geographical expansion and an increase in the prevalence of these parasites, or even the emergence of others. The presence of new parasite species underscores the importance of collaboration between the scientific community and the authorities responsible for hygiene and health controls in fish markets. Official control of fish-borne parasitic infections consists of a visual inspection of a few fish from each batch at the fish market by the veterinary services. However, this procedure cannot detect all infected fish. Thus, preventive measures are also based on consumer education. Knowledge of the survival of parasites along the entire food chain in fish products and reliable inactivation procedures are milestones for applying food safety principles. For these reasons, health authorities and governments should constantly inform consumers about the risks related to raw fish consumption. Therefore, our research findings can be used to create awareness among European sanitary authorities and translated into policies for managing fish-borne zoonotic parasites. Fish-borne zoonotic helminths are characterized by a complex epidemiology and life cycles, significantly influencing risks’ identification, prevention, and control. A clear comprehension of the life cycles of FBZHs is crucial to identify the potential source of infection and to design a risk map within the study area.

## 5. Conclusions

This study provides consistent epidemiological information on zoonotic helminths, describing their occurrence in freshwater fish populations of Italy and neighbouring countries, and highlighting potential risks to consumers. To date, data on the presence of zoonotic helminthic species and, in particular, seven FBHZs have been collected from wild fish species only and mostly in Italian water basins. Despite the fact that our findings predominantly reflect the Italian context, they highlight the need for further research in neighbouring countries to deepen our understanding of fish-borne zoonotic helminth occurrence and biology. The scientific community and health authorities should focus on expanding the parasitological survey where data are lacking. Defining outbreak areas and developing spatial databases will lead to target control efforts more efficiently and cost-effectively. The development of a joint European policy based on innovative strategies will help to fill the information gaps herein described. Providing actors throughout the food chain and sanitary authorities with reliable and updated information is critical for food safety management and decision-making. Educational programs are essential in establishing political participation in control programs at all societal levels and building consensus among the involved parties. Several improvements have been made for food-borne parasitic zoonoses, but with the current demographics and socioeconomic conditions, the occurrence of parasitic zoonoses will likely increase. In conclusion, we encourage a One-Health approach in the context of food safety among EU countries to manage the complex sanitary issues of all food-borne zoonoses.

## Data Availability

Data are contained within the article and Appendix A.

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
