# Peer review of "The Occurrence of Freshwater Fish-Borne Zoonotic Helminths in Italy and Neighbouring Countries: A Systematic Review"

_animals, 2023, doi:10.3390/ani13243793_

Round 1
Reviewer 1 Report
Comments and Suggestions for Authors
This paper studied the occurrence of freshwater fish-borne zoonotic helminths in Italy and bordering countries: a systematic review. FBZH is a common and major source of disease in freshwater fish. This article points out the types, sources, causes, and recommendations of the pathogen, which has certain reference significance. Overall, the writing is good and the research content of the topic has been successfully completed. English is also relatively fluent. There are a few small errors, the font size of the table is too large, and it spreads across pages. The genus name should be italicized. etc.
A few suggestions are as follows:
1. Line 198, The Clinostomum genus, the Genus name should be italicized
2. Table 1, the title is need to located at the top of the table
3. Line233, genus Contracaecum, the Genus name should be italicized. Revise the italicized generic names throughout the text.
4. Table 2,3,4,5, the title is need to located at the top of the table, and Put the table on one page of paper.
5. In 4. Discussion, There was only one paragraph in the discussion, and it was not well divided according to the viewpoints. It is suggested to modify it.
6. The conclusion section mainly discusses the significance and prospects of the article, but does not summarize the research results and conclusions of the article. It is recommended to supplement several main research conclusions.
Comments on the Quality of English Language
Carefully modify and refine sentence expression
Author Response
Dear reviewer, we thank you for your corrections and comments which have been accepted and allowed us to improve the quality of the paper. The font size of the table has been shrunk and the genus names have been italicized.
Line 198, The Clinostomum genus, the Genus name should be italicized
Line 198: Changed accordingly.
Table 1, the title is need to located at the top of the table
Line 237: Title of table 1 moved.
Line 233, genus Contracaecum, the Genus name should be italicized. Revise the italicized generic names throughout the text.
Changed accordingly. Genus names throughout the text have been italicized.
Table 2,3,4,5, the title is need to located at the top of the table, and Put the table on one page of paper.
Titles of the tables have been moved and table format has been reduced (only for one table it was not possible to fit in one page).
In 4. Discussion, There was only one paragraph in the discussion, and it was not well divided according to the viewpoints. It is suggested to modify it.
Section 4. Discusion: Thank you for your valuable comment, however we prefer to maintain the present format.
The conclusion section mainly discusses the significance and prospects of the article, but does not summarize the research results and conclusions of the article. It is recommended to supplement several main research conclusions.
Conclusion section: This section has been extended and the research results and conclusions of the article have been briefly added.

Reviewer 2 Report
Comments and Suggestions for Authors
In my opinion, the present paper could be accepted in “Animals” after a Minor revision.
Some minor comments are reported below.
Title: Why the authors included Germany, Croatia and Liechtenstein as bordering countries? Please explain or modify “bordering” to “other europeans”, considering that also in the MS (e.g., Line 30, 107, 123-124, 140).
Line 18: According to the instruction for the authors, the simple summary is mandatory in the structure of the MS, please provide.
Line 22: Please change “foodborne” to “food-borne” as reported in other parts of the MS.
Line 23: Please provide the full form for “FBZH” at the beginning of the sentence.
Lines 38 – 39: Please change the Keywords already present in the Title,
please change “Pseudaphistomum” to “Pseudamphistomum”.
I strongly suggest the authors to report the scientific names in italics in the Keywords and along the MS. Mainly in the result sections 3.2, 3.3 and 3.5.
Line 52: see Line 23.
Line 65: Please add “.” After “[13]”.
Line 71: Please delete “at” before “between”.
Lines 87: I suggest to remove “disease” before “progression”.
Line 143: The authors reported the infection indices as “(prevalence, abundance and intensity)”. In all tables included in the result sections they reported “P %, MI, A”, not in agreement with the captions. Please uniform that both in M&M and tables.
Line 169: There is a repetition, please delete one of them.
Line 179: Please change “.. commercial pellets as feed..” to “..commercial food..”.
Line 185: Please uniform the reference deleting “2020” before “[33]”, please check also in other parts of the MS.
Line 190: Please report the full name of the genus at the beginning of the sentence.
Line 198: Please check the scientific name.
Line 228: In table 1, as well as in other tables, please check the parasitological indexes of P%, MI and A. Delete the bold character in second line
Line 273: Please add “.” After “[273]”.
Line 297: Please change “(Cyclops)” to “(Cyclops)”.
Line 309: Please add “.” After “[90,95]”.
Line 322: Please delete “P” before “[95]”. Add “.” after the reference.
Line 356: Put the genus in italics
Line 389: Change “Ameirus” with “Ameiurus”
Table 3: Trasimeno lake Change I. melas in A. melas
Line 426 and 459: Please change “(Cyprinid)” to “(Cyprinid)”.
Line 496: I would like to suggest the authors to better report the data in the table because it is not so easy to read what they wrote in the first line
Line 535: Change Paramphistomum with Pseudamphistomum
Line 550: see Line 23.
Line 564: Please change “described herein” to “herein described”.
References: Please check all the scientific names reported in the reference list, focusing on the italics form.
Author Response
Dear reviewer, we thank you for your corrections and comments which have been accepted and allowed us to improve the quality of the paper.
Title: Why the authors included Germany, Croatia and Liechtenstein as bordering countries? Please explain or modify “bordering” to “other europeans”, considering that also in the MS (e.g., Line 30, 107, 123-124, 140).
Line 38, 114, 121, 131, 149, 608, and through the text: Thank you for noticing it, you are right. We Changed as suggested.
Line 18: According to the instruction for the authors, the simple summary is mandatory in the structure of the MS, please provide.
Line 18-25: Summary of the countries considered has been added.
Line 22: Please change “foodborne” to “food-borne” as reported in other parts of the MS.
Changed accordingly here and through the text.
Line 23: Please provide the full form for “FBZH” at the beginning of the sentence.
Line 31: The full form for “FBZH” has been added at the beginning of the sentence.
Lines 38 – 39: Please change the Keywords already present in the Title, please change “Pseudaphistomum” to “Pseudamphistomum”.
Line 46-47: Thank you for noticing it. Changed accordingly.
I strongly suggest the authors to report the scientific names in italics in the Keywords and along the MS. Mainly in the result sections 3.2, 3.3 and 3.5.
Line 46-47: Thank you for noticing it. Changed accordingly.
Line 52: see Line 23.
Line 59: Thank you for noticing it. The full form for “FBZH” has been added at the beginning of the sentence.
Line 65: Please add “.” After “[13]”.
Line 72: Added.
Line 71: Please delete “at” before “between”.
Line 79: Deleted.
Lines 87: I suggest to remove “disease” before “progression”.
Line 95: Suggestion accepted.
Line 143: The authors reported the infection indices as “(prevalence, abundance and intensity)”. In all tables included in the result sections they reported “P %, MI, A”, not in agreement with the captions. Please uniform that both in M&M and tables.
Line 152, 164 and tables: Indices have been uniformed in text and .
Line 169: There is a repetition, please delete one of .
Line 178: The title of figure 1 has been rephrased as follow: “The PRISMA flowchart reporting the literature search results and the screening process.
Line 179: Please change “.. commercial pellets as feed..” to “..commercial food..”.
Line 188: Thank you for your valuable comment, however we prefer use commercial pellets.
Line 185: Please uniform the reference deleting “2020” before “[33]”, please check also in other parts of the MS.
Changed accordingly and through the text.
Line 190: Please report the full name of the genus at the beginning of the sentence.
Line 199: Thank you for noticing it. The full of the genus has been added at the beginning of the sentence.
Line 198: Please check the scientific name.
Thank you for your valuable comment, done.
Line 228: In table 1, as well as in other tables, please check the parasitological indexes of P%, MI and A. Delete the bold character in second line
Tables: Parasitological indexes have been uniformed, thank you for the suggestion but we prefer to keep the second line in bold character.
Line 273: Please add “.” After “[34]”.
Line 284: Added.
Line 297: Please change “(Cyclops)” to “(Cyclops)”.
Line 309: Changed.
Line 309: Please add “.” After “[90,95]”.
Line 321: Added.
Line 322: Please delete “P” before “[95]”. Add “.” after the reference.
Line 334: Deleted and added.
Line 356: Put the genus in italics
Line 362: Changed accordingly.
Line 389: Change “Ameirus” with “Ameiurus”
Line 406: Changed accordingly.
Table 3: Trasimeno lake Change I. melas in A. melas
Table 3: Changed accordingly.
Line 426 and 459: Please change “(Cyprinid)” to “(Cyprinid)”.
Changed accordingly.
Line 496: I would like to suggest the authors to better report the data in the table because it is not so easy to read what they wrote in the first line
Thank you for your valuable comment, however, we prefer maintain the present format.
Line 535: Change Paramphistomum with Pseudamphistomum
Line 587: Changed accordingly.
Line 550: see Line 23.
Line 601: The full form for “FBZH” has been added at the beginning of the sentence.
Line 564: Please change “described herein” to “herein described”.
Line 616: Changed accordingly.
References: Please check all the scientific names reported in the reference list, focusing on the italics form.
Reference section: Italicized all the scientific names reported in the reference list.
